# Suggestions for Improvements in National Radon Control Strategies of Member States Which Were Developed as a Requirement of EU Directive 2013/59 EURATOM

**DOI:** 10.3390/ijerph19073805

**Published:** 2022-03-23

**Authors:** James P. Mc Laughlin, Jose-Luis Gutierrez-Villanueva, Tanja Perko

**Affiliations:** 1School of Physics, University College Dublin, 4 Dublin, Ireland; james.mclaughlin@ucd.ie; 2Radonova Laboratories AB, 751 38 Uppsala, Sweden; 3Institute for Environment, Health and Safety, SCK CEN & University of Antwerp, 2400 Mol, Belgium; tanja.perko@sckcen.be

**Keywords:** EURATOM BSS, radon action plans, building codes, risk communication

## Abstract

Exposure to the indoor air pollutant radon is considered to be a significant health risk globally, as has been demonstrated by many studies over time. A recent WHO statement on radon estimates that, worldwide, approximately 80,000 people may die every year due to lung cancer associated with radon exposure. The recent years have also seen huge improvements in radon policies in European countries, as a consequence of the issuing, in 2013, of the Council Directive 2013/59/Euratom. Although the protection of workers from radon exposure is well established, the protection of the general public needs more improvements. The main objective of this paper is, first, to acknowledge and recognise the improvements in radon protection policies, but also to show that there are many areas where improvements are desirable and possible. The final goal is to suggest better ways to protect the general population from exposure to radon gas. The suggestions are based on the experiences of the co-authors, who come from different disciplines related to radon management. The following fields or areas where improvements are possible are identified: risk communication, building codes, radon policies, including funding, research and protection of children. We describe the work that has been conducted, and the possible improvements and solutions in these fields.

## 1. Introduction

The naturally occurring radioactive gas radon is classified as a human carcinogen by the IARC, and the WHO considers that, after smoking, exposure to radon is a significant cause of lung cancer [1,2]. Due to its synergistic effects, radon exposure is an important lung cancer risk factor among smokers, who are at a 25 times higher risk of developing lung cancer from radon exposure than non-smokers [3]. The WHO also considers radon to be an important cause of lung cancer in people who have never smoked. Epidemiological studies of both occupationally exposed workers and members of the public exposed to radon in their homes have established quantitative evidence that prolonged exposure to radon, and its short-lived progeny, may significantly increase the risk of lung cancer [4,5,6]. Lung dosimetry studies support the epidemiological evidence [7].

Based on representative national radon surveys in 66 countries, including China and India, some estimates of radon-attributable deaths from lung cancer in the global population have been made [8]. Notwithstanding the many uncertainties involved in such estimations in this study, it was estimated that, globally, these deaths are likely to exceed 250,000 per year. On the other hand, the WHO recently estimated the annual global death rate from radon to be circa 80,000 [3]. Irrespective of the accuracy of such estimates, the global health burden from radon exposure represents a significant public health risk. To help address this problem, the EU Basic Safety Standards Directive has required MS to set a reference level no greater than 300 Bq/m^3^, while lower national reference levels of 100 or 200 Bq/m^3^ have been adopted in a number of countries, especially for dwellings (e.g., Ireland, the United Kingdom, Estonia, Denmark, Finland, Sweden, and the Netherlands) [9,10]. An important part of many national radon control strategies has been to encourage members of the public to measure radon in their homes, and to remediate them when the level exceeds the reference level [11]. In spite of the considerable efforts from radiation protection authorities over the past half century, there has been very little voluntary action from the public to deal with radon in their homes [12]. There is a growing opinion in the field of radon management that a more mandatory approach might be worth considering. It also seems that simply continuing to increase public knowledge about the hazards of radon exposure will not inevitably result in a significant number of voluntary actions being taken by the public to measure or reduce radon levels in their homes. Here, it is suggested that increasing emphasis should be placed on identifying gaps, or presently underused opportunities, in a number of relevant areas. From these gaps, additional tools to reduce public radon exposure could be developed. A few of these gaps have been identified, and are briefly described below. This is based on the experiences of the co-authors, who have expertise in different disciplines within the overall field of radon management. It should be noted that their listing sequence is not meant to imply any order of importance.

## 2. Some Gaps in Existing National Radon Control Strategies

### 2.1. Radon Risk Communication

There is a legal requirement in the EU member states to increase public awareness, and to inform local decision makers, employers and employees of the risks of radon, including in relation to smoking [11]. Moreover, the EU member states shall provide, as appropriate, approaches for the involvement of stakeholders in decisions regarding the development and implementation of strategies to manage exposure situations. According to this legal requirement, the EU member states are supposed to develop and implement radon awareness campaigns. Unfortunately, the content analysis of radon-related information on 173 internet pages, from national, regional and local radon policy actors, demonstrated that the availability of radon information on the internet pages in radon prone areas, is limited, that websites contain inconsistent radon information, and that the information is not supported by engaging stories. In addition, internet pages do not provide personalized features, or allow for stakeholder feedback and dialogue. Moreover, the use of social media for radon-related communication is often not in place, in order to improve test and remediation rates by residents under radon risk [13].

However, it is reported, in many social science studies, that the behaviour of people under radon risk is loaded with a value–action gap [14,15,16]. A value–action gap exists due to no, or low, correlation between the knowledge about radon (radon awareness) and actually doing a home radon test, or applying mitigation actions, in the affected population. As an example, the recent results of a public opinion study, conducted with respondents living in a high radon prone area in Belgium, demonstrate this gap; although 75% of the respondents stated that they are aware of radon, only 15% of them tested for radon, remediated if there was radon detected, or installed preliminary protective measures when the building was built [17]. 

This value–action gap indicates that radon management is not only a technical problem, but is also a socio-psychological problem, since residents under radon risk need to change their behaviour, test for radon, apply protective measures and remediate, if necessary. For this, communication about radon risk and behavioural recommendations to residents are critical challenges, and, unfortunately, are also the pitfalls for the responsible authorities [16,17,18]. Efforts to communicate the risks of radon and behavioural recommendations to avoid them have many gaps, and are limited in various aspects. Bouder et al. pointed out that recommended radon communication, as organised by authorities, is still often not evidence-based, theory-based nor strategic [11]. The results of a systematic review of health communication campaigns in the field of radon found that, although not recommended, an informative tone of voice (factual, scientific and numerical information) in radon communication prevails. Other components, such as emotional or social components, are often not included. Furthermore, the focus of radon risk communication is mostly on intention and less on behaviour itself, and on testing, instead of mitigation. There is also a huge lack of evaluation of how effective the radon communication campaigns were, and if people really changed their behaviour after the campaigns (Apers, Vandebosch, and Perko, upcoming) (Apers et al., upcoming).

### 2.2. Building Codes

Indoor radon levels are often incorrectly perceived to be natural. This is due, in part, to the way in which indoor radon levels are actually presented as natural in texts and on websites dealing with the radon problem. While radon itself is a naturally occurring radionuclide, the level of radon in any building is not natural, but is rather anthropogenic, as it is a consequence of the ways in which we site, design, construct and use a building. With present day building techniques in Europe, it is technically possible, and not difficult, to construct buildings in which the indoor radon levels would be below the existing reference levels. This is even the case in so-called high radon prone areas. Because of this, the drafting and modification of building codes are excellent examples of where there are opportunities to exercise an increased degree of control over indoor radon levels, both in the construction of new buildings and in the restructuring of existing buildings.

In the case of future homes, while some national building codes do include radon protective measures as design and construction requirements, in the majority of such codes there is not a post-construction inspection requirement to measure the radon level in the completed building [19,20]. In most countries where there is such a gap in the codes, such a requirement could, and should, be added to the building codes. Improvements in the codes may also be needed regarding the procedures used to verify the quality of the workmanship. It is important that the building codes contain a requirement that the quality of the installed radon protective measures should be inspected at the level of each individual house or building., In addition, there should, ideally, be a requirement that the indoor radon level of the completed home or building has to be measured within a year of it being occupied. Where either of these suggested requirements are absent, such gaps in the building codes should be filled.

As an example of the potential influence of building codes on indoor radon levels, extracts from the recently approved Spanish technical building code are now presented. The European Directive EURATOM BSS 59/2013 states, in Article 103, that “member states shall ensure that appropriate measures are in place to prevent radon ingress into new buildings. These measures may include specific requirements in national building codes” [10]. Spain published the modification of its technical building code in December 2019. The new Spanish building code [20] includes the requirement of the EU BSS, and has a new section dedicated to radon protection in new and existing buildings.

These new Spanish regulations apply to new buildings and existing buildings, where there are changes in the structure, aiming to introduce new sections in the building to change the building type or to reform the building. In the last case, the document clearly requests that the rehabilitation work should be considered as an opportunity to increase the existing radon protection.

The requirements for new buildings in Spain to adhere to the reference level of 300 Bq/m^3^ depends on the zone of the country where the building is constructed. Accordingly, the Spanish territory is divided into two zones, zone 1 and zone 2, and this classification covers more than 8000 municipalities. If a new building is designed for zone 1, it must include a radon barrier whose characteristics are defined in the building code. Those buildings planned to be built in zone 2 should incorporate, in addition to the radon barrier, an extra radon protection system. This is similar to the radon protocols of the Building Regulations of Ireland 1997 [19].

In cases where there are rehabilitation works in existing buildings, and the annual radon average is above the reference level, the Spanish building code indicates that the same radon protection measures as those for new buildings have to be applied, depending on how much the reference level is exceeded. The document also describes the procedure to determine the annual radon activity concentration. In particular, the radon measurements must be performed by a radon measuring service, accredited according to the ISO standard 17025 [21].

The above refers to the examples of Ireland and Spain, but there are also building codes in other EU member states. We have observed that, in many cases, there is a lack of information regarding indoor radon measurements in new buildings; either such a requirement does not exist in the code or, if it does, the writing is ambiguous.

### 2.3. Financial Tools

In many European countries, financial support for radon measurements, the construction of radon safe homes and the remediation of radon problem homes is limited, and could usefully be increased. Using health economics, such incentives can be shown to be positive for most European societies, from both public health and property value perspectives. 

The annex XVIII of the EURATOM BSS 2013/59 recommends, in Item 12, “where appropriate, provision of financial support for radon surveys and for remedial measures, in particular for private dwellings with very high radon concentrations.”

In Sweden, the government has carried out, for several years, a programme to facilitate radon remediation work on those buildings where radon concentrations were above the national reference level. The programme provided funding up to 50% of the total cost of the remediation work, with a maximum of EUR 2500.

The Swedish radon remediation funding was reintroduced on 1st July 2018. To make use of that funding, the user has to show that the annual reference level in the building has been measured according to the Swedish protocol, and by an ISO 17025 accredited radon measurement service. 

There is another issue related to the financial aspects of radon management. In many European countries, there seems to be a shortage of companies involved in the business of the remediation of radon problem houses. This is apparently because the low level of demand for such services has led to low profitability for remediation work, in spite of the fact that the market for radon remediation work in many countries is potentially large enough to make it financially attractive to both radon measurement and remediation companies. If there was more adequate state financial support available for radon measurement and remediation, it can be argued that this should, in principle, help to stimulate increased work opportunities for the radon companies, thereby increasing employment and profitability. In this context, it should be noted that companies involved in radon prevention work on new buildings, required by national building codes, can be quite profitable. The development of more professional and profitable radon companies is, and can be, greatly helped by courses in measurement and remediation. 

Banks and other financial institutions in Europe, which provide finance for home purchases or improvements, could be encouraged to include, in their mortgage agreements, a mandatory certification, or verification, that the radon concentration in a property is, or will be, at an acceptable level. In parts of the UK, a radon test is a legal requirement during the completion of conveyancing. A similar approach has been used for some decades in parts of the US, but, up to the present, has had limited use in European countries. As property rights laws in most countries are complex, adopting such a financial/legal approach to the radon problem in homes may be difficult to implement. Finally, it is also important to bear in mind the potential for conflict between radon retrofitting of radon problem houses and energy efficiency regulations. In some cases, strict adherence to energy efficiency requirements may actually cause an increase in radon levels. 

### 2.4. Protection of Children

At present, if an owner or occupant of a dwelling is made aware that the indoor radon concentration significantly exceeds that required by the EU BSS, she or he can, with impunity, choose not to remediate, or to carry out, further measurements. In keeping with the present zeitgeist, this “freedom to choose” attitude may be acceptable for adult occupants of a dwelling. If, however, there are children living in the dwelling, it raises the question of whether the right of the children to a healthy living environment is being infringed by the choices of the adults in the house. There are parallels, in this case, with radon exposure in schools, and protecting children from the direct and indirect effects of tobacco smoke.

When a new school is built, it must conform to national building codes regarding the control of radon (i.e., installation of barrier membranes, sumps, etc.). When the level of radon in a school is found to be above either the EU reference level of 300 Bq/m^3^, applicable to workplaces, or a lower national reference level, remediation should be mandatory. Fortunately for the children, any radon exposure protection actions carried out to protect the health of the workers, such as teachers and support staff in the school, will also coincidentally be of radiological health benefit to the children attending the school. Because of this, in Irish schools, in the interest of protecting children, an advisory or non-mandatory reference level of 200 Bq/m^3^, which is the same for homes, is used [19]. 

It is worthy of note that the parents of children attending a radon problem school usually welcome radon remediation measures, and, indeed, often lobby the relevant authorities for their implementation. In spite of this, parents are often not interested in testing for radon in their own homes. This behaviour is somewhat surprising, as their homes are often located close to the school on similar geology, and are commonly of a similar construction type, and are, therefore, likely to have a radon level similar to that of the local school. It should be noted, however, that the perception of the health risk to their children from radon in schools is sometimes considered of little importance by some parents [22]. 

Whether or not children are at a health risk due to elevated radon levels in schools or in their homes is not clear from a scientific perspective. There have, for example, been some ecological studies on the association between the exposure of children to radon and childhood leukaemia, but no association has been detected [23]. 

As lung cancer is not a disease of children, no epidemiological studies on a possible association between radon exposure and lung cancer in childhood have taken place, nor is it suggested here that they should be. Studies of underground miners exposed to the carcinogen radon have, however, shown that there can be a long latency period of many years between radon exposure and the appearance of lung cancer. Therefore, as a precaution, it seems prudent to try to develop strategies to protect children from high radon exposure in their homes. In addition, because of the biological sensitivity of developing children, protecting them from any form of air pollution has been long recognised [24]. In this regard, measures similar to those adopted in some countries to protect children from environmental tobacco smoke might usefully be considered.

### 2.5. Targeting Decision Makers

Radon case studies related to communication and stakeholder engagement, presented by Turcanu et al. [25], showed that there are more challenges related to the responsibilities, rules and roles of different actors involved in radon management, which may be barriers for the successful implementation of a radon mitigation strategy. Some of these challenges include the following: the plurality of responsibilities between different authorities; lack of awareness about radon and the associated risks among decision makers and the general public, as well as among key professional stakeholders, such as family doctors, architects, and building professionals; disparities of knowledge, with greater awareness of radon in the highest radon risk areas compared to other areas. 

While radon risk communication programmes usually target the public, increased targeting of decision makers and politicians at national and local government levels may be more effective. Such targeting of decision makers may be most effectively performed by lobbying by the radon industry and professional radon associations.

### 2.6. Research Needs

A systematic review of social science studies in the field of radon showed that there is a lack of social science, in general, and of comparative studies to support radon communication campaigns, in particular; although the attitudes and behaviours of sub-populations could differ from those of the general public, they are mostly not investigated and addressed; the methods and scales used in surveys should be improved; there is a lack of accurate selection and description of sampling strategies, measurement tools, research protocols and data analysis procedures; ethical aspects are often overlooked [26]. 

The new area of citizen science initiatives, the process whereby citizens are involved in science as researchers [27,28], can meaningfully contribute to national radon action plans, especially in the development of citizen science initiatives, which consider not only testing, but also radon mitigation, as well as research formulation. Martell et. all [29] identified eight past or ongoing citizen science initiatives related to radon from the following five countries: Canada, Ireland, Israel, Italy and the United States [29]. In radon research, there are emerging citizen science projects in the context of the EU H2020 project RadoNorm (www.radonorm.eu, accessed on 30 January 2022), where citizens have a role in knowledge production, data collection, formulation of research questions, interpreting data and disseminating scientific results.

Another important observation regarding research needs concerns issues arising when comparing dosimetric and epidemiological approaches to assess risks from radon exposure.

To calculate the doses to the lungs as a result of inhaling radon and its progeny, mathematical calculations are applied in dosimetry and biokinetic models, using accepted parameters, such as weighting factors for ionising particles, assumptions on the identification of sensitive cells in the bronchial epithelium, respiratory model parameters, etc. [5]. Irrespective of the current formally adopted values of some of the parameters used in lung dosimetry research, improving the accuracy of their estimations should be encouraged and supported. As the radiation doses to lung tissue, from the decay of radon and its deposited progeny in the lung, cannot be measured, the dosimetrically estimated lung doses from radon are mathematical constructs. These estimated doses are useful, if not essential, to assist regulatory authorities in implementing the requirements of the EURATOM BSS, in particular, for occupational exposure to radon. 

On the other hand, for the assessment of risk arising from exposure of the general public to radon, it seems more appropriate that robust evidence from future residential epidemiological studies, when available, should, from a scientific, rather than a purely regulatory perspective, take priority over dosimetric model-based risk assessments. Decisions on protecting the public could then be made on actual radon exposure data, rather than on estimated doses. Through the use of reference levels, this, effectively, is the current practice.

In the case of future radon epidemiological studies, the existence of possible synergistic effects between the exposure to radon and other indoor pollutants should be taken into account. Just as in the case of previous epidemiological studies, where obtaining data on smoking habits was, and will continue to be, an essential component of such studies, this would require obtaining exposure data on other indoor pollutants, in addition to that of radon. Another research area of importance that needs to be supported in future is the search for, and identification of (using techniques such as proteomics), biomarkers specific not just to ionising radiation as such, but also specific to alpha radiation of the lung or other tissues, which is the case with radon and progeny.

## 3. Conclusions

In many European countries, during the past half century, an important component of radon control strategies directed at the public has been to encourage and persuade the public to measure radon in their homes, and to remediate if considered necessary. For many reasons, related to poor communication interventions, scepticism, behavioural characteristics, financial considerations and general apathy, etc., voluntary public responses to these efforts have been disappointingly low. Here, we have tried to identify some gaps or elements in the existing radon control strategies that we believe need to be more fully exploited than heretofore. In identifying these gaps, we have also made some suggestions for improvements that might be considered, while recognising that the introduction of such changes may be difficult. As there has been little voluntary public action taken against exposure to indoor radon up to the present, a more mandatory approach in some gap areas may also be appropriate and effective.

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
