# Peer review of "Suggestions for Improvements in National Radon Control Strategies of Member States Which Were Developed as a Requirement of EU Directive 2013/59 EURATOM"

_ijerph, 2022, doi:10.3390/ijerph19073805_

Round 1
Reviewer 1 Report
Certainly, the authors’ opinion merit sharing with readers. However, it would be good the reader to find more fact-based suggestions – the facts are in clear deficiency in the text. In particular, I would stress on providing data about the numbers of radon tests and mitigated buildings in the countries in North America (USA, Canada), Europe and elsewhere – sometimes the facts and numbers might be the best speaker. I suggest major revision in which also the following points should be addressed:
- lines 48-49: Provide references for the countries listed – ref. [9] is not relevant in that sense;
- lines 85-87: Please specify separately the % (from the 15% tested) of remediated buildings;
- lines 123-124: Instead “In some countries…” I recommend to use “In countries like …(list the countries, with references)…”;
- Recommend along with comments about home purchases and improvement (lines 187-194) to comment on the potential of the energy-efficiency retrofits (supported by many governments) to raise radon levels. Number of publications in the last years report such increase in significant proportion of buildings that underwent such retrofit;
- lines 221-226: It is somewhat excessive to expect increased lung-cancer morbidity in childhood. This malignancy develops in more advanced age, after decade(s) of latency time. Suggest to comment that because of such latency time lung cancer induced by radon exposure in the childhood could develop in more advanced age;
- Section numbering should be corrected: 2., 2.1 (not 3.1) etc…;
- There are some typos that should be corrected: e.g. superscripts (m3) are needed on line 47-48, on line 276 “ae” should be corrected to “are” etc.
- Are refs. 19 and 21 different?
Author Response
Thanks so much for the review report. We appreciate your time reviewing our paper and you can see below our responses to your comments. We need to point out to the reviewer the obvious fact that their original line numbers do not now correspond to the final line numbers.
To see the new version of the manuscript please see the attachment
- lines 48-49: Provide references for the countries listed – ref. [9] is not relevant in that senses.
Done.
- lines 123-124: Instead “In some countries…” I recommend to use “In countries like …(list the countries, with references)…”;
New text:
“It is important that the building codes contain a requirement that the quality of the installed radon protective measures should be inspected at the level of each individual house or building, In addition there should ideally be a requirement that the indoor radon level of the completed home or building should be measured within a year of it being occupied. Where either of these suggested requirements are absent such gaps in the building codes should be filled.”
- Recommend along with comments about home purchases and improvement (lines 187-194) to comment on the potential of the energy-efficiency retrofits (supported by many governments) to raise radon levels. Number of publications in the last years report such increase in significant proportion of buildings that underwent such retrofit.
New text: “Finally it is also important to bear in mind the potential for conflict between radon retrofitting of radon problem houses and energy efficiency `regulations. In some cases strict adherence to energy efficiency requirements may actually cause an increase in radon levels.”
- lines 221-226: It is somewhat excessive to expect increased lung-cancer morbidity in childhood. This malignancy develops in more advanced age, after decade(s) of latency time. Suggest to comment that because of such latency time lung cancer induced by radon exposure in the childhood could develop in more advanced age
New text:
“As lung cancer is not a disease of children no epidemiological studies on a possible association between radon exposure and lung cancer in childhood have taken place nor is it suggested here that they should be. Studies of underground miners exposed to the carcinogen radon have however shown there can be a long latency period of many years between radon exposure and the appearance of lung cancer. Therefore as a precaution it seems prudent to try to develop strategies to protect children from high radon exposures in their homes. Here again there are parallels with protecting children from both active and passive smoking.”
- There are some typos that should be corrected: e.g. superscripts (m3) are needed on line 47-48, on line 276 “ae” should be corrected to “are” etc.
The incorrect appearance of m3 as m3 seems to be an unexpected feature of the word processing software. We have fixed it
- Are refs. 19 and 21 different?
Refs 19 and 21 are the same. We have reviewed the reference and adjusted accordingly.
- Section numbering should be corrected: 2., 2.1 (not 3.1) etc…; done
Reviewer 2 Report
The manuscript contributes to the discussion on how to obtain a more extensive protection from radon exposure, an important public health issue.
Some, although not all of, the possible elements affecting the effectiveness of the strategies of European Member States are reported and adequately discussed.
However, the paper could be significantly improved in some parts, as outlined below:
1) Some evaluation, also taken from literature, on the expected impact (on the rate/proportion of remediated houses, see also point 3 below) of the suggested improvements would be very valuable and useful and would make some proposals less generic, although acceptable.
2) In some parts, remedial actions to reduce radon concentration are considered only for levels above the Reference Level (RL), whereas (as also reported in the Directive 2013/59/Euratom) optimization has to be done also for levels below the (RL). In addition several papers have shown that, especially for a RL of 300 Bq/m3, reducing only levels above RL would have a small impact on the overall health impact of radon exposure.
3) In some parts of the manuscript a more extensive promotion from Member States is suggested, including financial support. However, this is suggested at the same level for both the measurements of radon concentration (which costs few tens euro) and the remedial actions to reduce the measured radon concentration (actions that costs hundreds or thousands euro). It is well known that the major problem is the low number/rate of remedial actions, much more than the number of measurements of radon concentration. Moreover, one of the authors is working for a company selling radon concentration measurements, so suggesting a financial support for radon concentration measurements (which would be still useful, although much less than financial supporting remedial actions) could be considered by some readers as an issue rising a conflict of interest. Therefore, it is recommended to considered in a differentiated way the need (and the suggestions) to increase the number of remedial actions and those to increase the number of radon concentration measurements, focusing, is possible, on the former.
Some minor (and more specific) point are reported below:
4) it is suggested to use "Council Directive 2013/59/Euratom" (e.g. lines 15-16)
5) Some references could be improved, e.g.:
there are more updated WHO-IARC reports than that one reported as Ref.1;
Ref.5 (which shuold be related to epidemiological studies) is largely focused on measurement issue and only partially to epidemiological studies, whereas reports of other international organizations (e.g. UNSCEAR) contains more specific reviews of epidemiological studies.
6) Lines 46-47: "EU Basic Safety Standards Directive has set a Reference Level 46 for public exposure to radon at 300 Bq/m3" should be corrected (e.g., with "EU Basic Safety Standards Directive has required MS to set a reference level not greater than 300 Bq/m3")
7) line 49: If possible, references to the RL of the countries should be added to the list of references.
8) line 107: Indoor radon levels are actually presented in texts and website as of natural origin, and this will probably contributes to the mentioned perception.
9) lines 155:157: some references could be added, if possible.
In conclusion, the authors are recommended to try, where it is possible, to improve their good manuscript on the basis of the above considerations.
Author Response
Thanks so much for the review report. We appreciate your time reviewing our paper and you can see below our responses to your comments. We need to point out to the reviewer the obvious fact that their original line numbers do not now correspond to the final line numbers.
To see the new version of the manuscript please see the attachment.
The manuscript contributes to the discussion on how to obtain a more extensive protection from radon exposure, an important public health issue.
Some, although not all of, the possible elements affecting the effectiveness of the strategies of European Member States are reported and adequately discussed. While the authors agree with these comments
However, the paper could be significantly improved in some parts, as outlined below:
- Some evaluation, also taken from literature, on the expected impact (on the rate/proportion of remediated houses, see also point 3 below) of the suggested improvements would be very valuable and useful and would make some proposals less generic, although acceptable.
This paper is an opinion piece primarily intended to identify what the authors perceive to be some gaps in current radon control strategies and also to help stimulate discussion of them in the radon community. The authors do agree with the comments of the referee that evaluations of the expected impact of the suggested improvements would be valuable. However we consider presenting such evaluations to be outside our intended scope for this opinion piece.and more appropriate to be given in detailed research papers by specialists in the relevant areas of expertise.
- In some parts, remedial actions to reduce radon concentration are considered only for levels above the Reference Level (RL), whereas (as also reported in the Directive 2013/59/Euratom) optimization has to be done also for levels below the (RL). In addition several papers have shown that, especially for a RL of 300 Bq/m3, reducing only levels above RL would have a small impact on the overall health impact of radon exposure.
We agree that “ reducing only levels above RL would have a small impact on the overall health impact of radon exposure”. A comment to this effect and optimisation below RL has been included in the revised manuscript.
- In some parts of the manuscript a more extensive promotion from Member States is suggested, including financial support. However, this is suggested at the same level for both the measurements of radon concentration (which costs few tens euro) and the remedial actions to reduce the measured radon concentration (actions that costs hundreds or thousands euro). It is well known that the major problem is the low number/rate of remedial actions, much more than the number of measurements of radon concentration. Moreover, one of the authors is working for a company selling radon concentration measurements, so suggesting a financial support for radon concentration measurements (which would be still useful, although much less than financial supporting remedial actions) could be considered by some readers as an issue rising a conflict of interest. Therefore, it is recommended to considered in a differentiated way the need (and the suggestions) to increase the number of remedial actions and those to increase the number of radon concentration measurements, focusing, is possible, on the former.
We will include the following text in the conflict of interest: “It should be noted that while one of the authors works for a company that offers a radon measurement service the company is not involved in promoting or providing radon prevention or remediation services”
Some minor (and more specific) point are reported below:
- it is suggested to use "Council Directive 2013/59/Euratom" (e.g. lines 15-16)
Correction done
5) Some references could be improved, e.g.:
there are more updated WHO-IARC reports than that one reported as Ref.1;
Ref 1 was included as it was historically a key monograph by WHO/IARC which for the first time “officially “identified radon as a Class 1 human carcinogen, We are aware of more recent WHO/IARC publications on radon and therefore include in the revised manuscript a reference to WHO/IARC Monograph Vol 78 Part 2 Ionizing Radiation : Some internally deposited radionuclides. (2001) . This excellent monograph contains extensive discussions on the risks from radon.
Ref.5 (which shuold be related to epidemiological studies) is largely focused on measurement issue and only partially to epidemiological studies, whereas reports of other international organizations (e.g. UNSCEAR) contains more specific reviews of epidemiological studies.
We agree with this comment and now additionally include in the revised text reference to the UNSCEAR 2019 Report.
6) Lines 46-47: "EU Basic Safety Standards Directive has set a Reference Level 46 for public exposure to radon at 300 Bq/m3" should be corrected (e.g., with "EU Basic Safety Standards Directive has required MS to set a reference level not greater than 300 Bq/m3")
Done
7) line 49: If possible, references to the RL of the countries should be added to the list of references.
We have used the WHO reference database which ,inter alia, contains a list of RLs adopted by various counties
8) line 107: Indoor radon levels are actually presented in texts and website as of natural origin, and this will probably contributes to the mentioned perception.
We agree. Accordingly we have changed modified the beginning of the Building Codes section to read : “Indoor radon levels are often incorrectly perceived to be natural. This is due in part to the way in which indoor radon levels are often actually presented as natural in texts and websites dealing with the radon problem. “
9) lines 155:157: some references could be added, if possible.
Since this is an opinion paper we have decided to avoid increasing the number of references.
Reviewer 3 Report
Please, find the attached file

Author Response
Thanks so much for the review report. We appreciate your time reviewing our paper and you can see below our responses to your comments. We need to point out to the reviewer the obvious fact that their original line numbers do not now correspond to the final line numbers.
To see the new version of the manuscript please see the attachment
- Line 37 . Please, adjust the parenthesis of the reference I do not understand what the referee wants to say
Done
- Since one of the aim of the paper is to identify areas “ where improvement is desirable and possible” I suggest to add some comments also regarding the lack of harmonization and identification of a unique methodology for the redaction of the radon maps. Indeed, in line 140-141 it is stated that ‘Accordingly , the Spanish territory is divided in two zones, zone 1 and zone 2 and this classification covers more than 8000 municipalities’. This map covers all the municipalities? if not, why? Maybe the following recent paper could better complete the description of the Spanish situation. Fernández, A.; Sainz, C.; Celaya, S.; Quindós, L.; Rábago, D.; Fuente, I. A New Methodology for Defining Radon Priority Areas in Spain. Int. J. Environ. Res. Public Health 2021, 18, 1352. https:// doi.org/10.3390/ijerph18031352
We agree with the referee that the lack of harmonization is an issue. However, to cover this subject is out of the scope of the present paper since it requires a more extensive explanation
- line 150-155 It is a very interesting point. This part could be also extended focusing on the fact that measurements could be performed not only by using passive systems but active, too. I think, in general, it should be done an endorsement on the use of active measurement systems. Moreover also standards as ISO 11665 should be mentioned and taken into account.
We agree with the referee but again the proposed explanation exceeds the intended length considered for the paper.
- line 174 Another issue to discuss at this point is that in some areas where radon is too high (for example in some Italian regions many houses easily overcome 200Bq/m3) another financial problem regards the fact that the real estate market is not very favourable to this approach since there could be very negative repercussions on the selling price.
Although we agree with the referee, it is a two-fold problem. If retrofitted homes with radon concentration levels previously above the reference could prove that they have been mitigated, this could potentially increase the value of the home.
Round 2
Reviewer 1 Report
The revision is OK for me.